



**Spatial-temporal distribution, photoreactivity and environmental**
**control of dissolved organic matter in the sea-surface microlayer of**
**the eastern marginal seas of China**
Lin Yang [a], Jing Zhang [a, c, *], Anja Engel[d], Gui-Peng Yang [a, b, c, *]
[a] Frontiers Science Center for Deep Ocean Multispheres and Earth System, and Key Laboratory of
Marine Chemistry Theory and Technology, Ministry of Education, Ocean University of China,
Qingdao 266100, China
[b] Laboratory for Marine Ecology and Environmental Science, Qingdao National Laboratory for
Marine Science and Technology, Qingdao 266237, China
[c] Institute of Marine Chemistry, Ocean University of China, Qingdao 266100, China
[d] GEOMAR Helmholtz Centre for Ocean Research, 24105 Kiel, Germany

[*] Corresponding authors. Key Laboratory of Marine Chemistry Theory and Technology, Ministry of Education, Ocean University of China, Qingdao 266100, China
*E-mail addresses*: zhangjouc@ouc.edu.cn (J. Zhang), gpyang@ouc.edu.cn (G.-P. Yang)





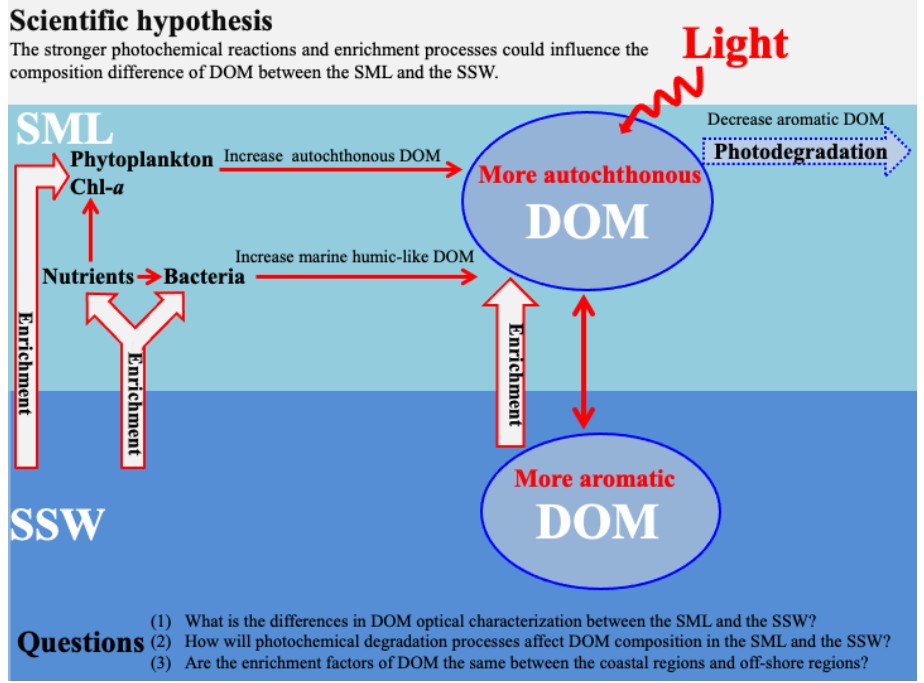

13          Graphical Abstract



**Abstract**
As the boundary interface between the atmosphere and ocean, the sea-surface microlayer (SML)
plays a significant role in the biogeochemical cycles of dissolved organic matter (DOM) and
macronutrients in marine environments. In our study, chromophoric DOM (CDOM), fluorescent DOM,
dissolved organic carbon, chlorophyll *a*, picoplankton, nutrients, and bacteria were frequently enriched
in the SML. We focus specifically on the optical properties in the SML, and we find that the
enrichment factors (EFs) of tryptophan-like component 4 was significantly higher than other
fluorescence components; the longer wavelength absorption values of CDOM showed higher EFs in
the SML, and the more significant relationship between CDOM and Chl-a in the SML, indicating that
autochthonous DOM was more frequently enriched in the SML than the terrestrial DOM. We find that
higher EFs were generally observed in the SML in the off-shore regions than in the coastal regions,
and CDOM in the SML is photobleached less after relatively strong irradiation, as also indicated by
the lower percentages of humic-like DOM and lower specific UV absorbance values ($SUVA_{254}$) in the
SML than the subsurface water (SSW). In combination with the SSW, the elevated nutrients may
stimulate phytoplankton growth, biological activity and then production of abundant fresh
autochthonous DOM in the SML. Our results revealed a general enrichment model and the more
autochthonous properties of DOM in the SML than the SSW for exploring the oceanic air-sea layer
environment.
**Keywords:** Sea-surface microlayer; Dissolved organic matter; Photochemical degradation;
Enrichment processes; Eastern marginal seas of China
**1 Introduction**





The sea-surface microlayer (SML) is the boundary interface between the atmosphere and the
ocean, which covers about 70% of the Earth's surface. SML is physicochemically distinct from
subsurface water (SSW, depth 3 ~ 5 m) and is characteristically enriched with phytoneuston,
chlorophyll, particulate carbon, dissolved organic matter (Hardy 1982; Hardy and Apts, 1989), and
biogenic organic compounds, such as lipids, proteins, and polysaccharides ((Liss and Duce, 1997; Liss
and Duce, 2005). With a total thickness ranging between 1 μm and 1000 μm, the SML remains present
in wind speeds of up to 6.6 m s$^{-1}$ (Wurl et al., 2011). A variety of processes contribute to the formation
of the SML in aquatic systems, these include but are not limited to, scavenging by rising bubbles,
atmospheric deposition, dissolved organic matter (DOM) photochemical degradation and
transformation, secretion, and biodegradation by organisms living within the microlayer (Neuston),
and migration of motile organisms into the SML (Aller et al., 2005; Wotton and Preston, 2005). The
role of the microlayer in oceanic emissions is not well understood and fundamental advance in
understanding its properties are needed. Because of its unique position at the air-sea interface, the
biological and photochemical reactions of DOM in the SML could strongly impact the biogeochemical
cycling of biologically important elements, for example, via the conversion of DOM into volatile
species such as carbonyl sulfide (OCS), which influence the atmospheric chemistry and climate
(Mopper et al., 2002). Air-sea gas exchanges of trace gases (e.g., CO, OCS, dimethylsulfide (DMS),
and alkyl nitrates gases) can also be greatly influenced by biological and photochemical reactions at
the sea surface (Blough, 1997).
Optical measurements of absorbance and fluorescence have been applied to track DOM
variability in aquatic ecosystems (McKnight et al., 2001; Zepp et al., 2004; Coble, 2007). The fraction
of DOM that absorbs light in the ultraviolet and visible ranges of the electromagnetic spectrum and the





fraction that exhibits a blue fluorescence are known as chromophoric DOM (CDOM) and fluorescence
DOM (FDOM), respectively, and their relative compositions can provide information differentiating
between autochthonous and allochthonous sources (Coble, 1996; McKnight et al., 2001; Stedmon et
al., 2007). Photolysis of DOM promotes the formation of low-molecular-weight compounds,
increasing the bioavailability of biologically refractory materials and facilitating carbon uptake by
microbes (Kieber et al., 1989). Indices based on optical measurements of absorbance and fluorescence
are commonly used to track DOM composition and infer DOM processing due to their low analytical
cost and high throughput relative to molecular level analyses (Coble, 2007; Fellman et al., 2010;
Gabor et al., 2014). Recent studies have mainly focused on using the characteristics of CDOM as
indicators of the sources and degradation states of DOM (Massicotte et al., 2017) in the SSW, and its
vertical distribution in estuaries and open oceans (Yamashita et al., 2017; Margolin et al., 2018).

Even though there are many studies that have documented the enrichment in DOM (e.g. amino

acids; carbohydrates) and inorganic nutrients in the SML relative to the SSW (Orellana et al., 2011;
Chen et al., 2016), the differences in organic matter composition between the SML and SSW, the
different enrichment factors of DOM in the SML between the coastal regions and the off-shore regions,
and how do photochemical degradation activities regulate DOM concentration in the SML need more
thorough discussion. Here, we investigated the concentration and composition of DOM in the SML
relative to the SSW and the responses of DOM to photoexposure. We hypothesized that the
photochemical reactions and enrichment processes could influence the composition difference of
DOM between the SML and the SSW, and greater solar exposure in the SML than in the SSW would
enhance the mineralization of DOM. To test these hypotheses, our study was designed to answer the
questions: (1) What is the differences in optical characterization of DOM between the SML and the





SSW? (2) Are the enrichment factors (EFs) of DOM the same between the coastal regions and
off-shore regions? (3) How will photochemical degradation processes affect DOM composition in the
SML and the SSW? We, therefore, compared the optical properties of DOM between the SSW and the
SML, and EFs of CDOM, FDOM components, dissolved organic carbon (DOC), chlorophyll-*a* (Chl-*a*),
nutrients, and bacterial abundance from the coastal waters to open ocean in the eastern marginal seas
of China (including the East China Sea (ECS) and the Yellow Sea (YS)) during spring of 2017 and
2019, summer of 2018, and winter of 2019; discuss how the composition of accumulated DOM was
affected by environmental conditions (wind speed and salinity) within the SML; and conducted
photoexposure experiments to compare photochemical degradation processes of DOM between the
SML and the SSW.

**2 Materials and methods**
*2.1* Study Area

Five cruises were conducted during the four seasons, specifically, from: 27 March to 15 April

2017 (R/V "*Dong Fang Hong 2*"), 26 June to 19 July 2018 (R/V "*Dong Fang Hong 2*"), March 2019
(R/V "*Zheyu No. 2*"), and 28 December 2019 to 16 January 2020 (R/V "*Dong Fang Hong 3*"). The
station locations are shown in Fig. S1. In spring, summer, and winter, SML samples were collected in
the YS and the ECS, which are shallow seas located almost entirely on the continental shelf in the
western Pacific Ocean where there is strong interaction between land and sea.
*2.2 Sampling*

We collected 220 paired SML and SSW water samples. SSW samples were collected at 2–5 m

depth    using    24    ×    10-L    Niskin    bottles    mounted    on    a    rosette    equipped    with    a



conductivity-temperature-depth (CTD) profiler. The SML samples were collected using the screen
sampling technique (Chen et al., 2016; Garrett, 1965) when conditions were calm. A screen sampler
with a 1.6 mm mesh of stainless steel wire on a 40 cm × 40 cm stainless steel frame was used. The
SML samples were collected in 500 mL brown sample bottles. The screen was held level and dipped
into the sea surface, moved laterally in order to sample from an undisturbed film, and then withdrawn
slowly from the surface. Repeated dipping was conducted until the desired volume was collected. The
screen method is often applied during field studies because of its relatively short sampling time and
large sample volume compared to other techniques (Momzikoff et al., 2004; Chen et al., 2016).
Immediately after collection, samples were filtered using 0.7 μm glass fiber filters (GF/F, Whatmann)
and the filtrates were transferred to 60 mL and 40 mL brown glass bottles (pre-cleaned and
pre-combusted) for later CDOM and DOC analyses. All samples were frozen (-20°C) and protected
from light, and upon arriving at the land laboratory, were analyzed as soon as possible.
*2.3 Photoexposure experiment*
SSW and SML water samples were collected in July 2018 at stations A3, BF, and H10 as well as
D2 and F6 located in the YS and the ECS, respectively. Samples (SSW: 2L; SML: 500 mL) were
immediately passed through 0.22 μm PES filters (Pall Corp. Port Washington, NY, USA) to remove
the majority of bacteria, placed in acid-washed and pre-combusted brown glass bottles and stored at
4°C. Similarly, filtered samples from each site were placed in five 80 mL optically transparent quartz
tubes (acid-washed and pre-combusted) and sealed without headspace or air bubbles to measure the
effect of light exposure. The quartz tubes were positioned on their sides under the irradiation source to
maximize the exposure of the sample; the water depth in each tube was 5 cm (i.e. the diameter of the
tube). Both sets (SML and SSW) were irradiated for 6, 12, 24, 50, and 88 h (25°C) in a GLZ-C



Quantum Sensor (Top Cloud-Agri Instrument, Zhejiang, China) solar simulator. All samples for DOC
concentration measurements were acidified to approximately pH 2.0 with high purity HCl and
analyzed within 7 d, and absorbance spectra and fluorescence excitation emission matrices (EEMs)
were run on non-acidified samples within 3 d of sampling (4°C and dark).
*2.4 Analytical measurements*
*Determination of the CDOM absorption coefficient*
Absorption spectra were determined using a UV-visible spectrophotometer (UV-2550 bi-channel;
Shimadzu, Tokyo, Japan) equipped with two 10 cm path-length quartz cuvettes. Sample absorbance
was automatically corrected for the absorbance of Milli-Q water. Absorbance scans ranged from 200 to
800 nm, with a spectral resolution of 1 nm. The absorption coefficient of CDOM was calculated
according to equation (1):

$a(\gamma) = 2.303A(\gamma)/l$                    (1)

where, $A(\lambda)$ is the absorbance at wavelength $\lambda$; and $r$ is the path length of the quartz cuvette in meters.
The spectral slope of the CDOM absorption curve (S) was calculated according to a non-linear
regression over the 275–295 nm and 350–400 nm wavelength range, according to:

$a(\lambda) = a(\lambda_0)exp[S(\lambda_0—\lambda)]+K$                    (2)

where, $\alpha(\lambda)$ is the absorption coefficient at wavelength $\lambda$; $\alpha(\lambda_0)$ is the absorption at the reference
wavelength $\lambda_0$ of 440 nm; S is the spectral slope; and K is a background parameter that accounts for
baseline shifts or attenuation due to factors other than CDOM. S was measured in the wavelength
ranges of 275–295 nm ($S_{275-295}$, $nm^{-1}$) and 350–400 nm ($S_{350-400}$, $nm^{-1}$). $S_{275-295}$ is used to characterize
DOM, with high values generally indicative of low-molecular-weight DOM that are linked to
photochemical modification (Helms et al., 2008; Ortega-Retuerta et al., 2009). The spectral slope ratio
($S_R$) was defined as the ratio of the two spectral slopes, $S_{275-295}$ to $S_{350-400}$. $S_R$ is also a sensitive
indicator of photochemically induced changes in the molecular weight within the CDOM pool, with





increases in $S_R$ suggesting stronger photochemical degradation (Helms et al., 2008; Ortega-Retuerta et
al., 2009). We used the absorption coefficient at 254 nm (a(254)) to determine the concentration and
distribution of CDOM in the SML from the eastern marginal seas of China. The specific UV
absorbance (SUVA$_{254}$) can be used to measure aromaticity (Weishaar et al., 2003) and molecular
weight (Chowdhury, 2013) of DOM, with higher values generally indicative of higher aromaticity.
*EEMs and determination of the CDOM fluorescence index*
EEMs were obtained using a F-4500 fluorescence spectrophotometer with a 1 cm quartz cuvette
(Shimadzu) (Hoge et al., 1993). The emission spectra were scanned every 5 nm from 250 nm to 550
nm, and at the excitation wavelengths between 200–400 nm at 5 nm intervals, with 5 nm slit widths
for the excitation and emission modes. The FL Toolbox, which was developed by Wade Sheldon
(University of Georgia) for MATLAB, was used to remove the Rayleigh and Raman scattering peaks
using the Delaunay triangulation method (Zepp et al., 2004). The fluorescence intensities of the
samples were corrected with Milli-Q water blank EEMs and then normalized to the water Raman
integrated area maximum fluorescence intensities (Ex/Em = 350 nm/365–430 nm, 5 nm bandpass)
(Coble et al., 1998; Singh et al., 2010). Raman units (RU) (Stedmon et al., 2007; Singh et al., 2010)
were used as the units for the Raman peak areas of water when the excitation wavelength of 350 nm
was used for correction. EEMs were modeled using PARAFAC in MATLAB 7.5 with the DOMFluor
toolbox (Stedmon and Bro, 2008).

$$X_{ijk} = \sum_{n=1}^{F} a_{in} b_{jn} c_{kn} + \varepsilon_{ijk}$$

(3)

where X$_{ijk}$ is the fluorescence intensity of the *i*th sample at the *k*th excitation and *j*th emission
wavelengths; $a_{in}$ is directly proportional to the concentration (scores) of the nth fluorophore in the *i*th
sample; $b_{jh}$ and $c_{kn}$ are the estimates of the emission and excitation spectra (loadings) of the nth
fluorophore at wavelengths *j* and *k*, respectively; *F* is the number of components (fluorophores); and
$\varepsilon_{ijk}$ represents the unexplained variability of the model (Singh et al., 2010). Split-half analysis
validation was used to determine the number of fluorescent components. The fluorescence intensity of
each fluorescent component was evaluated (Fig. S2, Supporting Information, Table 1).



*Determination of DOC, chlorophyll-a, heterotrophic bacterial abundance, dissolved oxygen, and*
*other parameters*
Concentrations of DOC were determined using the Shimadzu TOC-V$_{CPH}$ total organic carbon
analyzer with an injection volume of 80 μL. The accuracy of the test was ensured by measuring a deep
seawater reference (Hansell Laboratory, University of Miami) every 10 samples. The Chl-*a*
concentration was determined by a fluorescence spectrophotometer (7200-000, Turner Designs, CA)
after extraction in 90% acetone based on the procedure of Parsons et al. (1984). DO was determined
by iodination using the Winkler titration method (Carpenter, 1964), the endpoint was determined using
starch as a visual indicator. Salinity and temperature data were collected in situ by a
conductivity-temperature-depth sensor. All phytoplankton samples were enumerated in triplicate
according to Specification for Oceanographic Survey (State Bureau of Technical Supervision Bureau,
1992). Nutrient species concentrations were determined using an automatic analyzer (QuAAtro, Seal
Analytical, Germany) (Grasshoff et al., 2007). Heterotrophic bacterial abundances were measured by
flow cytometry (colorimetry,Beckman Coulter FC500-MPL) as described by Marie et al. (1997).
*Enrichment factors*
The enrichment factor (EF) in the SML is defined as follows
$$EF = C_M / C_S \tag{4}$$

where $C_M$ is the concentration of any substance in the SML; and $C_S$ is its concentration in the SSW. If
the EF of a substance is greater than 1.0, that substance is considered enriched, if it is less than 1.0, it
is considered depleted (Chen et al., 2016).
*2.5 Statistical analyses*





The correlation coefficient ($R$) and probability ($P$) values were used to evaluate the
goodness-of-fit. The correlation matrix, analysis of variance, and principal components analysis were
conducted with SPSS version 18.0 (SPSS Inc., Chicago, IL, USA) to determine the possible
relationships between the DOM parameters and environmental factors. A $P$-value $\leq$ 0.05 was
considered significant. Regression analyses between the optical parameters of DOM and several
biogeochemical parameters in the SSW and the SML samples were performed in the Table S1 and the
Table S2, respectively.
**3.Results and discussion**
*3.1 Distribution and chemical characterization of DOM in the SSW of the eastern marginal seas of*
*China*
The surface distributions of salinity, temperature, CDOM, DOC, Chl-*a*, and several optical
parameters in the study area during spring, summer and winter are shown in Fig. S3 (SSW)-S4 (SML)
(Supporting Information). There was a strong south-to-north temperature gradient, with warmer waters
in the ECS and cooler waters in the YS. Lower salinities were observed in the Changjiang Estuary and
coastal waters. The lowest mean wind speed was observed in the summer of 2018 (Table 2). In spring
and summer, the bacterial abundances were lower in the YS (spring mean concentration: $2.26 \times 10^8$
cells/L; summer mean concentration: $3.79 \times 10^8$ cells/L) than in the ECS (spring mean: $2.98 \times 10^8$
cells/L; summer mean: $7.64 \times 10^8$ cells/L), indicating that the warmer southern ECS had stronger
biological activity in the SSW.
The a(254) value ranged from 1.08 to 19.28 $m^{-1}$ in the SML and from 0.82 to 14.23 $m^{-1}$ in the
SSW during these three seasons. CDOM absorption values and DOC concentrations were generally
decreased from the inshore to the offshore stations (Fig. S3 c)-d)). Higher a(254) values were



generally observed in the Changjiang Estuary (spring: station D1 (4.13 m$^{-1}$); summer: station D2 (3.98
m$^{-1}$); winter: station D1 (3.14 m$^{-1}$)) and in the northern YS (spring: station A2 (4.26 m$^{-1}$); summer:
station H11 (5.37 m$^{-1}$); winter: station H12 (5.95 m$^{-1}$) ). There were significantly negative linear
correlations between salinity and a(254) in all cruises in the SSW (p < 0.01, Fig. 2), especially in the
ECS, implying that freshwater run-off and seawater mixing played a more important role in
determining CDOM distributions in the SSW. The strongest negative linear relationship observed
between salinity and a(254) was observed in winter when the influence of terrestrial input in this study
region was maximal. In addition, SUVA$_{254}$ ranged from 0.51 to 8.39 L mg C$^{-1}$ m$^{-1}$ in the SML. In
comparison with the SML, the SSW exhibited lower variability in SUVA$_{254}$ values from 0.63 to 5.39 L
mg C$^{-1}$ m$^{-1}$, with higher values at the northern YS stations and Changjiang Estuary coastal stations (Fig.
S3k)). According to the SUVA$_{254}$ trends observed by Massicotte et al. (2017), the DOM composition
we observed in the SSW of the Changjiang Estuary ecosystem were more similar to the DOM
measured in freshwater ecosystems than in the ocean. SUVA$_{254}$ underwent a sharp decrease from the
Changjiang Estuary ecosystem to the southeastern ECS, suggesting that aromatic and/or highly
conjugated DOM moieties were degraded along the aquatic continuum from the Changjiang Estuary to
the open ocean. Higher S$_{275-295}$ values were also observed in some off-shore stations (Fig. S4i)). These
comparisons showed that the DOM pools of the Changjiang Estuary contained molecules that were
more HMW-DOM and contained more aromatic compounds, CDOM in the SSW of the southeastern
ECS, which was derived predominantly from an autochthonous origin (phytoplankton production and
bacterial activity), clearly showed the presence of organic matter freshly released into sea (Yang et al.,
2020). The detail of mixing behavior, biological and photolytic degradation of dissolved organic
matter in the East China Sea and the Yellow Sea were discussed in our previous paper (Yang et al.,



2020).


*3.2 Fluorescence signature and factors controlling the composition of FDOM components in the SSW*
*and the SML*

FDOM properties can be used as the sensitive indicator of DOM processing and water mass. Four

fluorescent components were identified by PARAFAC analysis with the DOM Fluor toolbox in
MATLAB 7.5 (Stedmon and Bro, 2008), hereafter named C1, C2, C3, and C4 (Fig. S2). The
humic-like C1 and C3 were categorized as two traditional types of humic-like fluorescent components
(Coble 1996). Component 1 had primary fluorescence excitation and emission peaks at 345 nm and
455 nm, respectively, which was similar to terrestrial humic-like fluorophores in the visible region
(peak C) (Osburn et al., 2012). Relative to C1, the fluorescence of C3 was blue-shifted and had
fluorescence peaks at 385 nm emission and 315 nm excitation. The microbial humic-like component
had a relatively shorter emission peak wavelength compared to the terrestrial humic-like PARAFAC
components previously identified in the open ocean (Catala et al., 2015). C2 exhibited Ex/Em maxima
at 255 nm/310 (375) nm, which could be considered tyrosine-like fluorescence (Stedmon et al., 2003)
and attributed to autochthonous and/or microbially consumed FDOM. C4 had an excitation range of
280 nm with an emission peak at 335 nm, which corresponded to peak T of the amino-acid-like
fluorescence of tryptophan, likely derived from in situ primary autochthonous substances and other
fresh biological sources (Coble, 1996). The tryptophan-like C4 and the humic-like C1 and C3 in the
SSW were all negatively correlated with salinity ($P < 0.01$, Table S1), but increased with the
increasing DO level. These suggested that water mixing and microbial activity were important factors
in determining geographical distributions of FDOM in the SSW (Breitburg, et al., 2018; Yamashita et





al., 2017; Galgani and Engel, 2016). Moreover, the geographical distribution of humic-like C1 and
protein-like components were more similar to that of the Chl-a concentration in the SML (Fig. 3 a, b,
d). Such relationships suggested that the production of protein-like and humic-like FDOM with
phytoplankton production and decay in the SML.

FDOM enrichment in the SML of all stations ranged between 0.5 and 11 (n = 225) and FDOM

was more frequently enriched (C1: 89.6%; C2: 73.2%; C3: 91.8%; C4: 93.4% of all samples) than
CDOM. The fluorescence intensity of the components in the SML samples decreased in the following
order: tryptophan-like > tyrosine-like > terrestrial humic-like > marine humic-like; whereas those in
the SSW samples decreased in the order: tyrosine-like > tryptophan-like > marine humic-like >
terrestrial humic-like. The tryptophan-like component (C4) was mostly enriched in the SML samples
with a median EF = 2.2 and a range from 0.2 and 8.0. The EF of C4 was clearly higher than other
components in all seasons (Fig. 5b)), especially in summer, and the FDOM composition in the SML
revealed a relatively higher proportion of autochthonous tryptophan-like FDOM than the SSW. It has
also been broadly recognized that tryptophan-like C4 in the particulate fraction is related to recent
primary production (Brym et al., 2014; Yamashita, 2014) and that phytoplankton excrete
tryptophan-like fluorophores (Romare-Castillo et al., 2010). Together, as already emphasized
previously, the variation observed for FDOM can be more related to that of Chl-a in the SML, these
observations suggested that the DOM enriched in the SML was made up of a relatively higher
proportion of marine autochthonous DOM than the SSW.

*3.3 DOM and biogenic molecules accumulation in the SML*

Up to 88% of our CDOM samples were enriched in the SML, with the median EF for a(254) of

1.3, ranging between 0.4 and 6.7. Concentrations of CDOM, FDOM, nutrients, bacterial abundance,



and Chl-*a* in the SML were correlated with their respective SSW concentrations (Fig. 4),
demonstrating that transport from the SSW to the SML is an important pathway. Furthermore, the
relatively higher CDOM absorption enrichment value in the SML were found at longer wavelengths
(Fig. 5a)) EF of a(355) > EF of a(254)). Marine production of DOM had the largest influence on the
CDOM absorption properties in the longer wavelength range (Danhiez et al., 2017) ($S_{320-412}$: DOM
marine origin VS. $S_{275-295}$: terrestrial DOM). Galgani and Engel (2016) also observed that amino
acid-like fluorophores were highly enriched in the SML, not only due to their amphiphilic properties,
but also due to their local production in the SML. Therefore, the marine local production might
significantly affect the composition of DOM in the SML. Additionally, the nutrients showed
significantly higher EFs ($NO_3^-$: 3.41 ± 6.08, $n = 41$; $NO_2^-$: 3.57 ± 5.54, $n = 52$; $PO_4^{3-}$: 2.13 ± 2.74, $n =$
68; and $SiO_3^{2-}$: 6.53 ± 13.67, $n = 13$) than biological and DOM parameters in the SML. The strong
correlation between the SML and SSW concentrations of $NO_2^-$, $NO_3^-$, and $SiO_3^{2-}$ (Fig. 4) showed that
the similar fundamental drivers are probably at work in both compartments for these nutrients. For
example, dissolved substances, particles, and microorganisms were brought to the interface by simple
diffusion, rising bubbles (Jarvis, 1967), convection, and upwelling from sediments and subsurface
water, and at the same time, the microlayer is also a sink for fallout from the atmosphere (Duce et al.,
1976). In addition, we also observed the significant positive relationship between a(254) and Chl-*a* (*R*
= 0.662, *P* < 0.01) in the SML during spring, and the positive relationship between the EF of $PO_4^{3-}$ and
the EF of Chl-*a* (*R* = 0.319, *P* = 0.01, Table 3). These observations indicated that spatial variation of
CDOM concentrations were related to Chl-a in the SML. The enrichment of inorganic nutrients would
be an important factor influencing the production and composition of phytoplankton-produced DOM
(Carlson and Hansell, 2003) in the SML. Therefore, phytoplankton growth, primary productivity rate,





biological activity and marine autochthonous DOM production would all be enhanced by the enriched
nutrients in the SML.

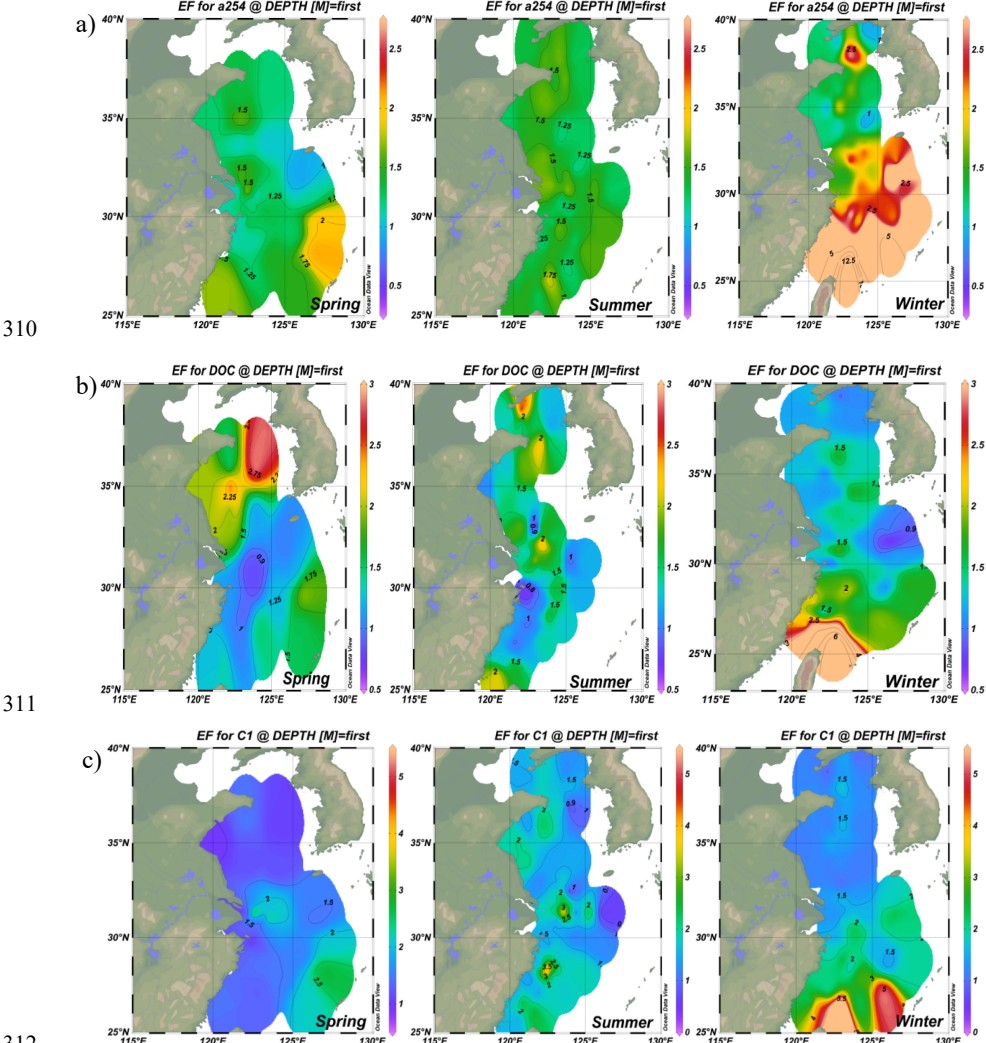










Fig. 1. Distributions of enrichment factors of CDOM, DOC, Chl-*a*, and four fluorescence components
in the surface microlayer water during spring, summer, and winter. Increasing DOM yields were
significant in coastal regions in all seasons, but the higher EFs were more pronounced in off-shore





regions.

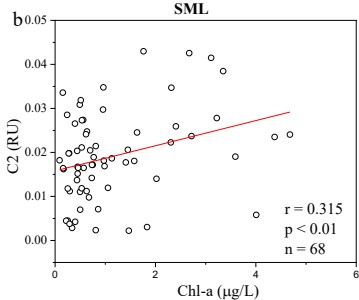



Fig. 2. Relationships between a(254) and salinity in the SSW in the YS and the ECS during spring,
summer and winter.






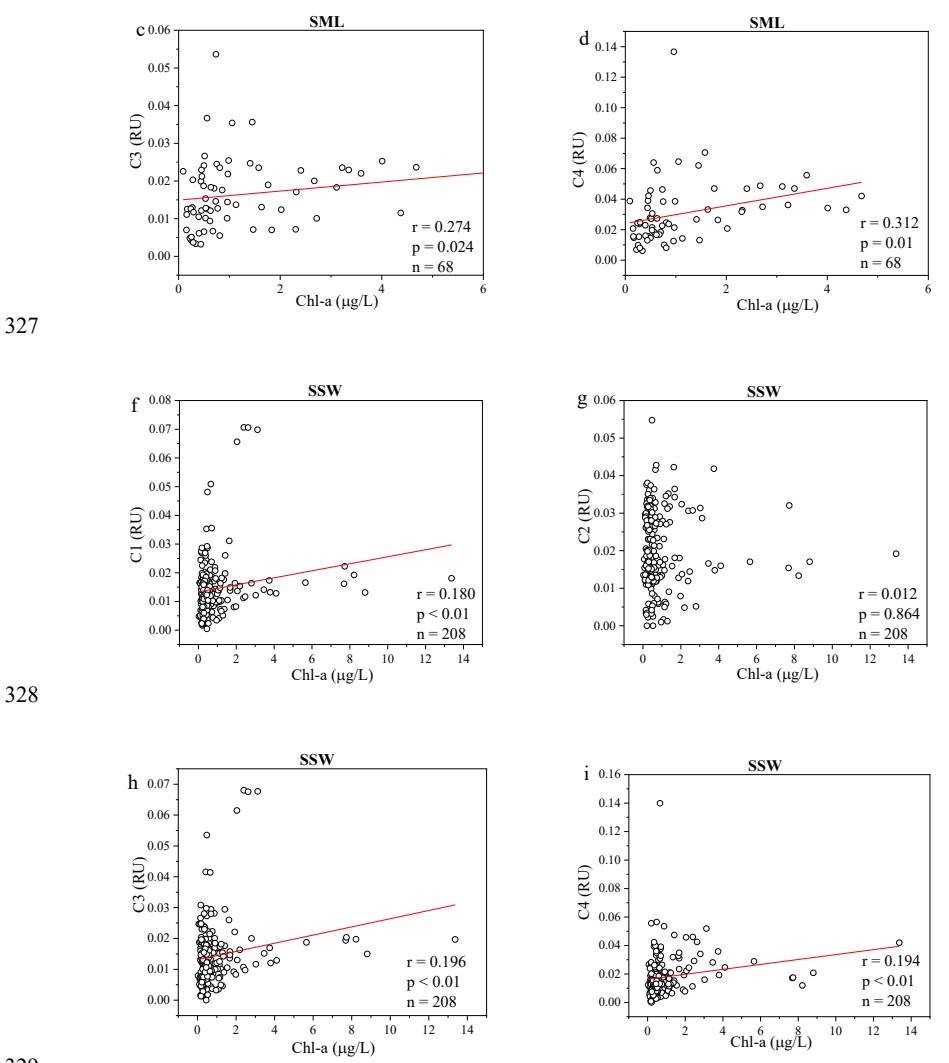




Fig. 3. Relationships between a(254), four fluorescence components and Chl-*a* in the SML (a-d) and in
the SSW (f-i).






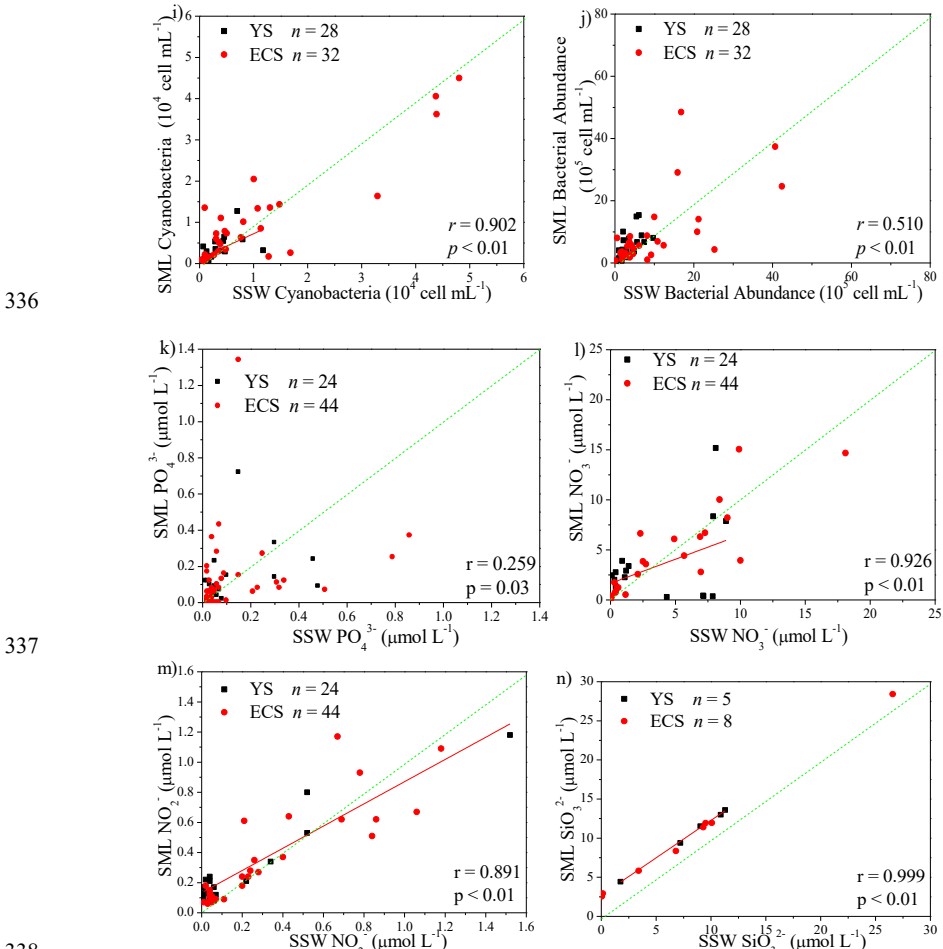

Fig. 4. Correlations between the microlayer CDOM, DOC, Chl-*a*, four fluorescence components
concentrations, cyanobacteria, phytoplankton biomass, nutrients and bacterial abundance, and their
subsurface water concentrations. The dashed lines correspond to the 1:1 lines, and the full lines are the
regression models. (All DOM spectroscopic parameters sample were analyzed in spring, summer,
autumn, and winter; Chl-*a* was determined in spring, summer, and summer; cyanobacteria,
phytoplankton biomass, nutrients and bacterial abundance were determined in spring and summer.).



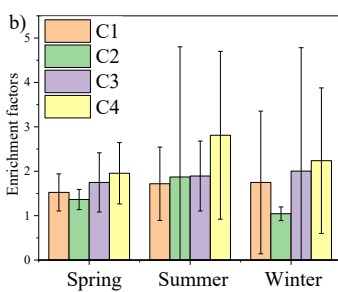

Fig. 5. Mean enrichment factor of $a_{CDOM}$ (254 nm and 355 nm), DOC, Chl-$a$, nutrients ($PO_4^{3-}$; $NO_3^-$; $NO_2^-$, $SiO_3^{2-}$), and four fluorescence components during spring, summer, and winter.

*3.4 Wind speed influencing the enrichment of DOM optical properties*

The wind speeds during our observations ranged from 0.2 to 14.9 m s$^{-1}$. We divided them into three different wind regimes: low (0.0–2.0 m s$^{-1}$), moderate (2.0–10.0 m s$^{-1}$), and high (10.0–14.9 m s$^{-1}$). Although the EFs of DOC and Chl-$a$ were negatively correlated with wind speed (DOC: $P = 0.002$; Chl-$a$: $P = 0.042$), the EFs of CDOM and FDOM were not. During the low wind regime, no significant relationships were apparent between wind speed and either EFs of CDOM or FDOM, CDOM and FDOM were consistently enriched, with EFs ranging from 1.0 to 2.2, and a mean a(254) EF value of 1.3 (n = 20). However, the EFs during moderate winds had larger variability and ranged from 0.9 to 14.5, with a mean EF value of 1.6 (n = 143), and during high winds they ranged from 0.6 to 1.8, with a lower mean EF value of 1.1 (n = 18). In addition, depleted levels of CDOM (EF < 1) occurred at frequencies of 5.6%, 9.1%, and 20.0% during low, moderate, and high wind regimes, respectively. Therefore, although lower wind speeds and ascending bubbles might further promote the transportation of organic materials from the underlying waters, DOM enrichments were still observed at wind speeds up to > 10 m s$^{-1}$. Reinthaler et al. (2008) also reported that higher enrichment was found at higher wind speeds. During moderate to high wind regimes, breaking waves not only can





disrupt the surface film and physically drive DOM back into the bulk water, but also facilitate the
formation of the SML as rising bubble plumes transported DOM to the surface, resulting in wider
ranges of EFs (Frew et al., 2004). Higher wind speed does enhance mixing (Reinthaler et al., 2008),
which can arguably favour transport of nutrients and DOM from the SSW equally (Wurl et al., 2011).
Although wind speed appear to play an important role in the enrichment of surface-active DOM, the
chemical composition of the SML influence its stability. For example, enrichments of sulphate
half-ester groups in the SML (Wurl and Holmes, 2008) could increase stability because these groups
can influence the intrinsic viscosity of marine polymers (Nichols et al., 2005) and sulphur-containing
algal carbohydrates are less soluble and hydrolysable (Kok et al., 2000). Enrichment processes and
biochemical processes of organic substances in the marine environment are all likely to be the more
important contributors of DOM to the SML in our study regions.
*3.5 Photochemical degradation of DOM in the SSW and the SML*

Photobleaching is one of the major mechanisms determining the geographical distributions of

chromophoric and fluorescent DOM in the ocean (Helms et al., 2008; Brinkmann et al., 2003; Siegel
et al., 2005). The average $SUVA_{254}$ values in SSW were generally higher than those in the SML in our
study regions (SSW: 2.45 ± 0.91 L mg-C$^{-1}$ m$^{-1}$ vs. SML: 2.39 ± 1.34 L mg-C$^{-1}$ m$^{-1}$), and the most
obvious distinction happened in  summer (Table  2).  These  indicated  that  although  CDOM
concentration in the SSW was lower than that in the SML, CDOM in the SSW has a higher degree of
aromaticity compared to the SML. Thus we performed photochemical incubation experiments to
confirm whether photochemical reactions influenced the differentiated aromaticity and photo-reactive
features of DOM between the SML and the SSW.

After 88 h of exposure, the a(254) values were only 49.6%, 45.5%, 42.1%, 41.8% and 37.0% of



the initial values at stations A3, BF, D2, F6, and H10 in the SSW, and 72.5%, 42.4%, 42.6%, 49.0%
and 44.0% of the initial values at stations A3, BF, D2, F6, and H10 in the SML, respectively. Overall,
a(254) and $SUVA_{254}$ decreased by $49.9 \pm 12.8\%$ and $43.0 \pm 15.5\%$, respectively, in the SML, and by
$56.8 \pm 4.7\%$ and $56.0 \pm 10.2\%$, respectively in the SSW. Therefore, stimulated solar UV exposure
caused a larger decrease in DOM absorbance in the SSW than the SML (Fig. 6). The relatively rapid
decrease of $SUVA_{254}$ in the SSW indicated a more rapid conversion of DOM to less humic-type
materials than in the SML. Approximately 65% of FDOM was lost during the irradiation experiment,
except in the case of the tyrosine-like component 2 from the SML at the station H10, which increased
slightly. Photoproduction of tyrosine-like components has been previously reported by Zhu et al.
(2017), who suggested that the photochemical degradation of CDOM contributed to the release small
amounts of tyrosine-like fluorophores. The tryptophan-like C4, humic-like C1 and C3 were more
photodegradated in the SSW than in the SML (Fig. 6c), e), f)). For example, C1, C3 and C4 show a
marked decrease in the SSW at the off-shore station F6. Because of the origin of CDOM at the station
F6 remote from the direct terrestrial influence, the majority of CDOM at the station F6 was thought to
be a by-product of net primary production. The present results showed that a large fraction of the total
CDOM in the SSW at the off-shore station F6 is still potentially sensitive to photooxidation. As
already referred previously, CDOM in the SSW showed higher $SUVA_{254}$ values, and higher
percentages of humic-like DOM than in the SML. Therefore, the photochemically mediated shifts in
DOM in the SSW were more pronounced than those in the SML in our incubation experiments, in
terms of both absorption and fluorescence values.

This heterogeneity in the EFs and photochemical reactivities of FDOM components can be

related to the chemical and structural nature, such as molecular weight, aromaticity or humification of



FDOM enrichment processes. Hydrophilic, carboxylic acid-bearing DOM moieties are preferentially
degraded by simulated sunlight (Brinkmann et al., 2003). The largest fractions of photolabile DOM are
made up of aromatic carbon rings or high double bond equivalent molecules (Kujawinski et al., 2004;
Gonsior et al., 2009). The humic-like C1 and C3, all of which exhibited significantly positive
relationships with $SUVA_{254}$ (< 0.001, Table 1) and shown higher aromaticity, were more prone to
photochemical degradation (Fig. 6c), e), and f)). The tyrosine-like C2, as compared to other
protein-like compounds, is generally considered more labile and susceptible to bacterial cycling and
rapid consumption by microbiota (Medeiros et al., 2015). The SML experiences the most intense solar
radiation, especially ultraviolet (UV) light (Obernosterer et al., 2005). Photochemical degradation may,
therefore, be a sink for aromaticity fluorescent components in the SML, and be a source for the
tyrosine-like C2. In addition, Blough (1997) discovered that photochemical production rates in the
SML should lead to the more rapid oxidative turnover of materials at the interface and potentially to
reactions and processes not observed in bulk waters. Therefore, differences in $SUVA_{254}$ values and
photoreaction behavior between the SSW and the SML may also reflect that DOM in the SML was
already photobleached, which resulted in the decrease of DOM aromaticity, CDOM in the SSW
appeared to be more susceptible to photochemical degradation than CDOM in the SML. Together,
photo-irradiation have the significant influence on the accumulation of protein-like DOM and
depletion of aromatic organic compounds in the SML, and organic carbon might have undergone a
more rapid cycling in the SML than the SSW.



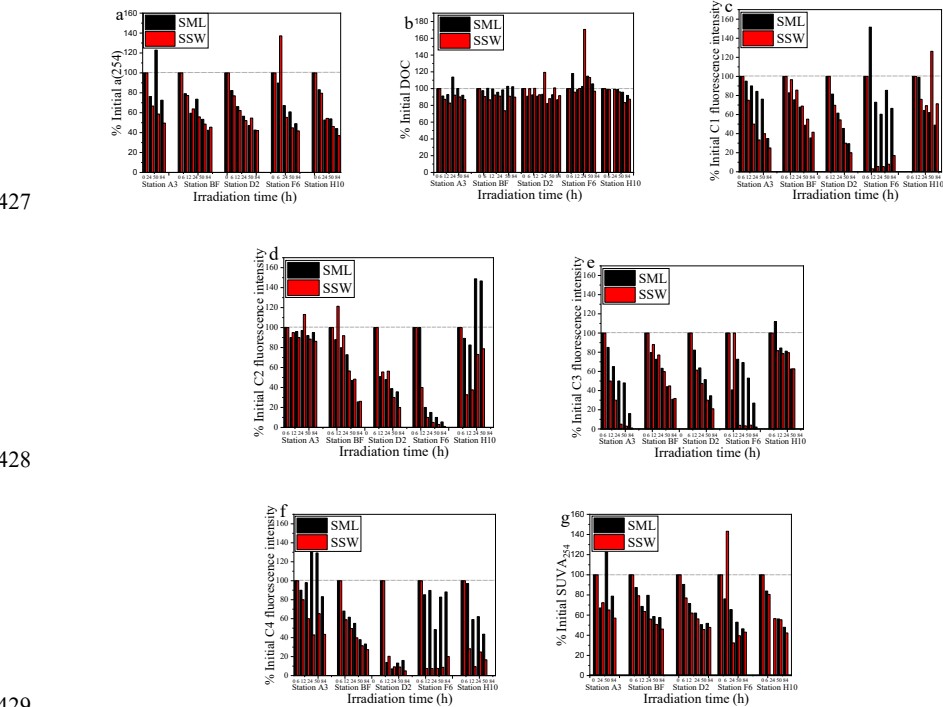




Fig. 6. Changes in ratios of a(254), DOC, SUVA$_{254}$ and four fluorescence components intensities
to initial values for both SML and SSW sample.


*3.6 Variations in the enrichment factors of CDOM, DOC, FDOM along the coastal regions to ocean*

The concentrations of a(254) and DOC decreased from the coastal regions to the open ocean, and

decreased from the northern part of the sampling area (the YS) to the southern part of the sampling
area (the ECS) in both the SSW and the SML (Fig. S3 c)-d) and Fig. S4 a)-b)). However, CDOM and
FDOM were more frequently enriched in the ECS (CDOM: 93% of all samples; FDOM: 72–94% of
all samples) than that in the YS (CDOM: 86% of all samples; FDOM: 70–92% of all samples). The
higher EF values for CDOM, FDOM, DOC, Chl-*a*, nutrients, and cell were generally observed in the
ECS (Fig. 1). Lower EFs and EFs < 1, which indicate a depletion of CDOM in the SML, were usually
observed at short distances from the coast (Fig. 1) with lower salinity. The salinity during our





observations ranged from 23.6 to 35.1. Although CDOM and FDOM concentration negatively
correlated with salinity, the EFs of CDOM and FDOM were weakly positive related with salinity (Fig.
7). The EFs of Chl-*a* and nutrient were also higher in the southeastern ECS (Fig. 1 and Fig. S5), where
sufficient light and higher temperature combined to facilitate primary production and higher
contributions of autochthonous materials to DOM. DOM in the SSW of the southern ECS was more
dominated by marine autochthonous materials in our previous discussion (Yang et al., 2020). The
Changjiang River discharges enormous amounts of N and P into the ECS (Liu et al., 2018), but
phosphorus is generally the major limiting element for phytoplankton growth in the ECS (Liu et al.,
2016). Thus the difference in EFs of CDOM and Chl-a between YS and the ECS, and between the
coast and off-shore regions is likely due to the significantly nutrients enrichment in the SML in the
off-shore regions. In winter, we observed especially higher EF values for CDOM and FDOM in the
southern ECS (Fig. 1a)-f)). With wind from the northwest (Weng et al., 2011), biologically essential
trace elements and anthropogenic emissions are carried from the land and can enter the ocean via the
SML by wet or dry deposition. The EFs of humic-like C1 and C3 were relatively high in winter (Fig.
5b)), probably due to the input from atmospheric deposition during winter, and the relatively low
CDOM concentrations in bulk water. Atmospheric deposition of organic carbon and nutrients were
found a peak in winter over the coastal ECS (Wang et al., 2019). We suggested that the EFs of CDOM
and FDOM increased from the coastal regions to the open ocean, and increased from the YS to the
ECS were likely due to the enrichment of enough nutrients in the SML in the open ocean promote
phytoplankton biomass and DOM production.





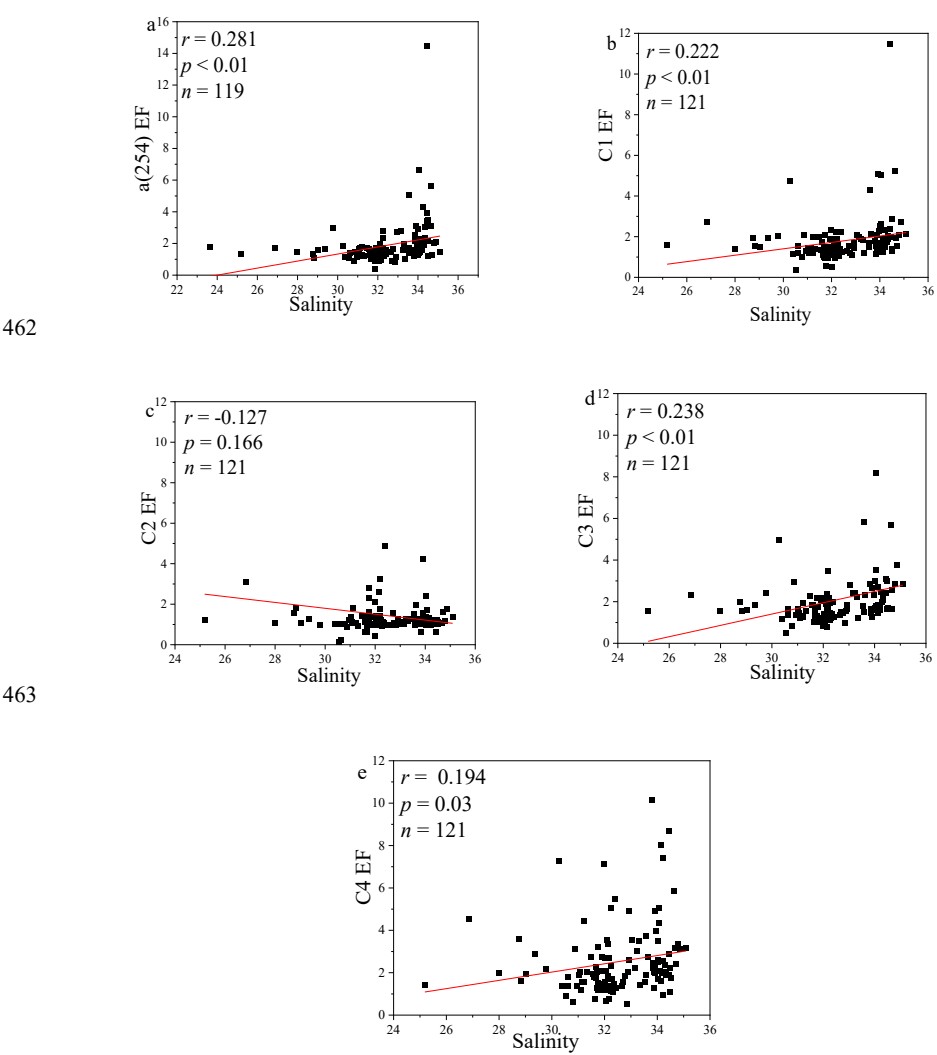

Fig. 7. Relationships between salinity and EFs of a(254), Chl-a, DOC, and four fluorescence components.



The SML is an aggregate-enriched biofilm environment with distinct microbial communities,
where the diversity of microorganisms can differ significantly from those of underlying waters (Liss
and Duce, 2005; Cunliffe et al., 2013), the heterotrophic bacterial abundance in the SML was ~ 7.5
fold greater than those in the SSW in the ECS during our spring cruise (Sun et al., 2020). Here, EFs
showed a greater presence of bacteria and marine protein-like DOM in the SML than that in the SSW,
while the protein-like DOM was linked to microbial utilization and degraded faster than the humic-like
substances (Yang et al., 2017; Jørgensen et al., 2011). Therefore, compared to coastal waters that have
larger terrestrial DOM and nutrients inputs, CDOM shown higher EFs in off-shore regions where
DOM in the SSW is mostly of marine autochthonous origin with higher temperatures and stronger
biological activity. The significantly higher abundance of cells, phytoplankton, nutrients, and
protein-like DOM in the SML supported microbial activities, and further contributing to the local
release of marine extracellular DOM directly from microbes in the SML in the off-shore regions.
When exposed to higher light intensities (summer), obviously enhance mineralization of DOM in the
SML, and relatively less photochemical degradation in SSW could result in lower percentage of
aromatic DOM in SML than the SSW. We concluded that SML CDOM dynamics can be expressed as
a simple balance among enrichment process, primary production and photochemical destruction. Thus,
higher EF of DOM in the SML in off-shore regions are likely supported by a favorable combination of:
1) deposition and accumulation of amphiphilic compounds, 2) importance of bubble for upward
transport of DOM and enrichment in SML, and 3) new production of DOM within the SML as a
consequence of higher nutrients enrichment and the primary production.

**4. Conclusions**



This study has provided the first data set that considers the distributions of CDOM, FDOM, DOC,
Chl-*a*, nutrients, and bacterial abundances in the SML and SSW of the ECS and the YS during spring,
summer, and winter. We have observed that the CDOM distribution related variability in primary
production in the SML. Furthermore, we have demonstrated that localized and stronger photochemical
oxidation may be responsible for the decrease in the aromaticity of the DOM in the SML, due to
enhanced transformation or removal of terrestrial DOM, compared with the SSW. We also
demonstrated that in off-shore seawaters away from terrigenous influence, the EFs of CDOM, DOC,
FDOM and Chl-a in SML tend to be higher in off-shore regions than those in coastal regions, because
of the relatively higher enrichment of nutrients which could enhance phytoplankton growth and
promoted plant production and DOM production in the SML. Multiple observations of spatial
distributions, seasonal variations, chemical compositions, and photochemical reactions of CDOM in
the SML have supported the hypothesis that stronger enrichment and photochemical processes occur in
the SML in ocean, resulting in relatively accelerated enrichment of more marine local production
DOM in the SML than the SSW.

**Acknowledgements**
We thank the captain and crews of the R/V 'Dong Fang Hong 2', the R/V 'Dong Fang Hong 3',
and the R/V 'Zhe Yu 2' for their assistance and cooperation during the investigation. We gratefully
acknowledge Ya-Hui Gao (Xiamen University), Yu Xin (Ocean University of China), and Xiao-Hua
Zhang (Ocean University of China) for providing the Chl-*a*, nutrients, bacterial abundance and
picoplankton data, respectively. This work was financially supported by the National Key Research
and Development Program (Grant No. 2016YFA0601304), and the National Natural Science
Foundation of China (Grant Nos. 41806093 and 491830534).

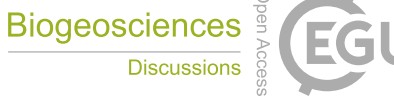

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





**Table 1** Spectral characteristics of the four fluorescent components identified by the PARAFAC

modal in this study, compared with those preciously identified.

| Component | $Ex_{max}$ (nm) | $Em_{max}$ (nm) | Coble (1996) | Comparison with other studies using PARAFAC | Description and probable source |
|---|---|---|---|---|---|
| C1 | 345 | 455 | peak C 320-360/420-480 | Osburn et al. (2012) | Terrestrial-like humic substances |
| C2 | 255 | 375 (310) | peak A 230-260/380-460 | Stedmon et al. (2003) | Tyrosine-like substances |
| C3 | 315 | 385 | peak M 290-310/370-420 | Stedmon and Markager (2005) | Marine humic-like substances (biological degradation) |
| C4 | 280 | 335 | peak T 270-280/340-350 | Coble (1996) | Tryptophan-like; Non-Humic-like; Biological production in the water column |





**Table 2** Average temperature, salinity, wind speed, CDOM a(254), DOC, Chlorophyll-*a* (Chl-*a*), dissolved oxygen (DO), $S_{275-295}$, $S_R$, and $SUVA_{254}$ of the SSW and SML in the BS, YS and ECS during spring, summer, autumn, and winter.

| | | Spring | | Summer | | Autumn | | Winter | |
|---|---|---|---|---|---|---|---|---|---|
| | Water layer | mean | SD | mean | SD | mean | SD | mean | SD |
| Temperature (ºC) | SSW | 14.0 | 4.91 | 24.0 | 3.66 | 13.7 | 2.69 | 14.0 | 5.23 |
| Salinity | SSW | 32.5 | 1.92 | 31.7 | 2.17 | 30.7 | 1.10 | 32.7 | 1.41 |
| Wind Speed (m s$^{-1}$) | SSW | 5.98 | 2.86 | 5.47 | 2.51 | 7.30 | 3.82 | 6.09 | 2.52 |
| DO (mg L$^{-1}$) | SSW | 6.44 | 0.85 | 7.57 | 1.07 | 7.49 | 0.51 | 8.32 | 0.99 |
| Chl-*a* (µg L$^{-1}$) | SSW | 1.26 | 2.38 | 1.13 | 1.48 | 0.74 | 0.36 | 0.42 | 0.25 |
| | SML | 1.63 | 3.66 | 1.28 | 1.13 | 0.61 | 0.29 | no data | |
| DOC (µmol L$^{-1}$) | SSW | 91.3 | 25.7 | 109.4 | 33.55 | 151.2 | 83.8 | 88.4 | 22.51 |
| | SML | 132.9 | 77.4 | 145.7 | 49.8 | 146 | 33.5 | 131.3 | 91.1 |
| a(254) (m$^{-1}$) | SSW | 3.20 | 2.49 | 3.10 | 1.34 | 5.21 | 1.84 | 2.52 | 1.26 |
| | SML | 3.70 | 1.98 | 4.05 | 1.66 | 5.51 | 1.82 | 4.74 | 2.50 |
| $S_{275-295}$ (nm$^{-1}$) | SSW | 0.0201 | 0.0049 | 0.0188 | 0.0035 | 0.0214 | 0.0075 | 0.0207 | 0.0068 |
| | SML | 0.0222 | 0.0073 | 0.0178 | 0.0021 | 0.0207 | 0.0067 | 0.021 | 0.0055 |
| $S_R$ | SSW | 1.723 | 1.026 | 1.731 | 1.557 | 1.493 | 1.312 | 1.521 | 0.52 |
| | SML | 1.095 | 0.218 | 1.361 | 0.296 | 1.357 | 0.772 | 1.416 | 0.214 |
| $SUVA_{254}$ (L mg-C$^{-1}$ m$^{-1}$) | SSW | 2.067 | 0.664 | 2.244 | 0.671 | 3.008 | 0.949 | 3.008 | 0.949 |
| | SML | 1.911 | 0.768 | 1.951 | 0.359 | 3.196 | 1.126 | 2.992 | 1.034 |



**Table 3** Correlation coefficients between EF of DOM optical properties, Chl-$a$, DOC, $PO_4^{3-}$, $NO_3^-$,

$NO_2^-$, $SiO_3^{2-}$, Cyanobacteria, Picophytoplankton, Bacterial abundance.

| | EF of a(254) | EF of DOC | EF of Chl-$a$ | Ef of C1 | EF of C2 | EF of C3 | EF of C4 | EF of $PO_4^{3-}$ | EF of $NO_3^-$ | EF of $NO_2^-$ | EF of $SiO_3^{2-}$ | EF of Cyanobacteria | EF of picophytoplankton |
|---|---|---|---|---|---|---|---|---|---|---|---|---|---|
| EF of DOC | 0.185 | | | | | | | | | | | | |
| EF of Chl-$a$ | 0.092 | 0.021 | | | | | | | | | | | |
| Ef of C1 | **.336**\*\* | 0.047 | -0.119 | | | | | | | | | | |
| EF of C2 | 0.163 | 0.073 | -0.017 | **.635**\*\* | | | | | | | | | |
| EF of C3 | **.413**\*\* | 0.179 | -0.096 | **.907**\*\* | **.557**\*\* | | | | | | | | |
| EF of C4 | **.319**\*\* | 0.021 | 0.011 | **.574**\*\* | **.368**\*\* | **.628**\*\* | | | | | | | |
| EF of $PO_4^{3-}$ | 0.129 | 0.267 | **.319**\* | 0.131 | -0.037 | 0.139 | 0.087 | | | | | | |
| EF of $NO_3^-$ | -0.065 | -0.054 | 0.235 | -0.037 | -0.044 | -0.027 | -0.053 | 0.26 | | | | | |
| EF of $NO_2^-$ | 0.15 | 0.208 | **.307**\* | 0.142 | -0.017 | 0.192 | 0.035 | **.571**\*\* | 0.271 | | | | |
| EF of $SiO_3^{2-}$ | **.634**\* | 0.004 | 0.074 | 0.122 | -0.101 | 0.305 | 0.151 | 0.205 | -0.118 | -0.141 | | | |
| EF of Cyanobacteria | 0.091 | -0.017 | 0.028 | -0.027 | -0.105 | -0.047 | -0.052 | 0.218 | 0.027 | **.755**\*\* | -0.286 | | |
| EF of Picophytoplankton | **.347**\*\* | 0.0281 | 0.252 | -0.067 | -0.082 | -0.077 | 0.025 | 0.218 | -0.08 | 0.113 | -0.327 | 0.081 | |
| EF of Bacterial abundance | -0.036 | -0.069 | -0.063 | -0.061 | 0.004 | 0.014 | -0.093 | -0.12 | -0.026 | -0.073 | -0.099 | **.730**\*\* | -0.064 |

\*\* Correlation is significant at the 0.01 level (two-tailed)

\* Correlation is significant at the 0.05 level (two-tailed)