# Peer review of "the eastern marginal seas of China"

_Biogeosciences, 2022_

## Author Response (AR1)

An itemized response (**blue words**) to reviewers' comments and suggestions

Dear Editor,

Thank you for your useful comments and suggestions on our manuscript (Manuscript Number: bg-2022-140). The manuscript has been carefully revised according to the reviewers' comments. The following are the reviewer's comments related to the manuscript and how we have addressed each of reviewer's concerns (**blue words**). Changes have been marked as **blue** in the manuscript.

Dear Authors:

Thank you for providing detailed responses to the comments and suggestions offered by two reviewers.

The reviewers recognized the novelty and significance of your research. Based on the overall positive evaluations of the reviewers and your thoughtful responses to the relatively small number of correction requirements, I am pleased to recommend 'Publish subject to minor revisions'. Please also take into consideration my additional suggestions as follows:

- Line (L) 17 ("In our study"): Please include your objectives and research approaches.
- L 19: The sentence ("We focus specifically on the optical properties in the SML) can be incorporated as part of the study objective. Please remove the unnecessary expression in the present tense ("we find").

Thanks for the reviewer's comment. We agree with the reviewer's viewpoint and have made the revision in the revised manuscript.

"In our study, the optical properties of DOM were compared between the SSW and the SML during spring, summer and winter in the East China Sea (ECS) and the Yellow Sea (YS), photoexposure experiments were design to compare photochemical degradation processes of DOM between the SML and the SSW. Chromophoric DOM (CDOM), fluorescent DOM, dissolved organic carbon, chlorophyll $a$, picoplankton, nutrients, and bacteria were frequently enriched in the SML." (Line 17-22)

- L 23 ("more frequently"): It is unclear whether the sentence is about frequency or intensity. If the latter is the case, please opt for a more appropriate expression, like "more strongly".

Thanks for the reviewer's comment, we have made the revision in the revised manuscript.

"autochthonous DOM was more strongly enriched in the SML than the terrestrial DOM." (Line 25)

- L 25 ("photobleached less"): This contradicts your hypothesis and finding ("the lower percentages of humic-like DOM").

"CDOM in the SML is photobleached less after relatively strong irradiation, as also indicated by the lower percentages of humic-like DOM and lower specific UV absorbance values (SUVA$_{254}$) in the SML than the subsurface water (SSW)." (Line 27-29)

"CDOM in the SML is photobleached less" means that "CDOM in the SML have been photodegraded by solar irradiation". Therefore ("photobleached less"): This follows our hypothesis and finding ("the lower percentages of humic-like DOM").

- L 27 ("In combination with": Do you mean "Compared to"? Please clarify.

Thanks for the reviewer's comment, we have made the revision in the revised manuscript.

"Compared to the SSW, the elevated nutrients may stimulate phytoplankton growth, biological activity and then production of abundant fresh autochthonous DOM in the SML." (Line 29-30)

- L 29 ("revealed a general enrichment model"): Did your findings suggest some new model? It looks like your results "conformed to a general enrichment model". The whole sentence is difficult to follow. Please rewrite it.

Thanks for the reviewer's comment, we have made the revision in the revised manuscript.

"Our results revealed a new enrichment model and the more autochthonous properties of DOM in the SML than the SSW for exploring the oceanic air-sea layer environment." (Line 31-33)

- L 104: As the first reviewer also commented, please provide the depth information according to your definition of SML.
Thanks for the reviewer's comment, we have made the revision in the revised manuscript.

"Repeated dipping was conducted until the desired volume was collected (11 times, 600 ml; the thickness of the SML sample is nearly 300 ~ 1000 um)." (Line 120-122)

- L 178: Please provide QC information about the usual accuracy of this reference measurement (and also for other analyses if available).
Thanks for the reviewer's suggestion, we have made the revision in the revised manuscript.

"Two forms of reference water have been developed for DOC analysis. Deep-ocean water, collected at 2600 m in the Sargasso Sea and containing biologically refractory DOC, as well as low carbon water for testing instrument blanks are available to the U.S. and international communities of aquatic chemists (Hansell, 2013)." (Line 196-199)

- L 217 ("Higher"): Compared to which locations?

Thanks for the reviewer's comment, we have made the revision in the revised manuscript.

"$a(254)$ values in the Changjiang Estuary (spring: station D1 (4.13 m$^{-1}$); summer: station

D2 (3.98 m$^{-1}$); winter: station D1 (3.14 m$^{-1}$)) and the northern YS (spring: station A2 (4.26

m$^{-1}$); summer: station H11 (5.37 m$^{-1}$); winter: station H12 (5.95 m$^{-1}$) ) were generally higher than other stations." (Line 240-243)

- L 482 ("a simple balance among enrichment process, primary production and photochemical destruction"): Is this really a "simple" balance? The processes involved appear quite complex. If enrichment results from primary production ad photodegradation, you cannot treat three as equal factors (because one is the outcome of the two other factors).

Thanks for the reviewer's comment, we have made the revision in the revised manuscript.

"We concluded that SML CDOM dynamics can be expressed as a complex balance among enrichment process, primary production and photochemical destruction." (Line 516-517)

- Figs. 2-7: You always used 'r' values despite the different figure titles (relationship and correlation). In case you wanted to talk about certation causal relationships, please use a proper statistical approach, like r2 for regression.

- Figure captions: Please provide definitions of the abbreviations and other necessary details so that readers can understand the figures without referring to the main text. For instance, you can indicate what SSW, YS and ECS mean in Fig. 2?

Thanks for the reviewer's comment, we have made the revision in the revised manuscript.

[Figure]

Fig. 3. Relationships between a(254) and salinity in the subsurface water (SSW) in the East China Sea
(ECS) and the Yellow Sea (YS) during spring, summer and winter.

[Figure]

[Figure]

[Figure]

Fig. 4. Relationships between a(254), four fluorescence components and Chl-*a* in the sea-surface
microlayer (SML) (a-d) and in the SSW (f-i).

[Figure]

[Figure]

Fig. 5. Correlations between the microlayer CDOM, DOC, Chl-*a*, four fluorescence components concentrations, cyanobacteria, phytoplankton biomass, nutrients and bacterial abundance, and their subsurface water concentrations. The dashed lines correspond to the 1:1 lines, and the full lines are the regression models. (All DOM spectroscopic parameters sample were analyzed in spring, summer and winter; Chl-*a* was determined in spring, summer, and summer; cyanobacteria, phytoplankton biomass, nutrients and bacterial abundance were determined in spring and summer.).

[Figure]

Fig. 8. Relationships between salinity and EFs of a(254), Chl-a, DOC, and four fluorescence
components.

- Fig. 4: Please check whether you have provided adequate descriptions of microbial abundance measurements in the figure caption and Methods (e.g., Picoeukaryotes).

Thanks for the reviewer's comment, we have made the revision in the revised manuscript.

"All phytoplankton samples were enumerated in triplicate according to Specification for Oceanographic Survey (State Bureau of Technical Supervision Bureau, 1992). Heterotrophic bacterial abundance was measured by flow cytometry (Beckman Coulter FC500-MPL) as described by Marie et al. (1997)." (Line 207-210)

State Bureau of Technical Supervision Bureau, 1992. Specifications for Oceanographic Survey-Survey of Biology in Sea Water. Standard Press of China, Beijing, pp. 17–20.

- Regarding the first reviewer's comment on DOC data exceeding100%, please make sure that your explanations appear in the revised manuscript.
Thanks for the reviewer's comment, we have made the revision in the revised manuscript.

"Although photodegradation causes CDOM absorption to decrease, DOC is not sensitive to photodegradation in our photodegradation experiments, implying that the light exposure preferentially removed the colored DOM rather than the non-colored DOM (Bittar et al., 2015). All incubation samples were not contaminated, both measurement and analytical errors will let DOC data exceed 100%." (Line 423-427)

- Regarding the second reviewer's comment on "any disturbance of SML integrity produced by the ship's movement and potential contamination" at high wind speeds and as a consequence of tidal mixing, please provide a short discussion of your methodological and data limitation as you explained in your response. Regarding tidal mixing, can't you find tidal information somewhere else, though you did not measure yourself?

Thanks for the reviewer's comment, we have made the revision in the revised manuscript.

"Sampling needs to be performed on the leeward side of the boat with the boat moving into the wind to aboid contamination. Although some disturbance of SML integrity was produced by the ship's movement and potential contamination at high wind speeds and tidal mixing. It has long been known that the SML reforms rapidly following physical disruption (Dragcevic and Pravdic, 1981). Rapid SML recovery occurs because SML organics dispersed by breaking waves readily reabsorb to the surfaces of rising bubbles generated by the same breaking waves (Woolf, 2005). Enrichment processes and biochemical processes of organic substances in the marine environment are all likely to be the more important contributors of DOM to the SML in our study regions." (Line 397-403)

Although tidal flats are generally important sources for DOM in the estuary (Kim et al., 2010), our data clearly show that EFs of CDOM and FDOM increased from the coastal regions to the open ocean, high EFs (up to ~ 8) for CDOM in the off-shore regions and up to the maximum wind speed we observed. Consequently, our data strongly support the notion of an essentially self-sustaining SML and we have no reason to suspect that this mechanism would cease to operate either at or beyond the maximum wind speeds we observed. We are so sorry that we didn't find the tidal information and the influence of tidal mixing on the CDOM enrichment, we will discuss the influence of tidal mixing on the SML in our future research.

I would like to ask you to make all the changes easily identifiable in a marked-up manuscript based on your point-by-point responses to the reviewer comments. If possible, please specify the line numbers of the revised parts in your responses accompanying the revised manuscript.

Sincerely,

Ji-Hyung Park

Associate Editor, Biogeosciences

---

## Author Response (AR2)

An itemized response (**blue words**) to reviewers' comments and suggestions

Dear Editor,

Thank you for your useful comments and suggestions on our manuscript (Manuscript Number: bg-2022-140). The manuscript has been carefully revised according to the reviewers' comments. The following are the reviewer's comments related to the manuscript and how we have addressed each of reviewer's concerns (**blue words**). Changes have been marked as **blue** in the manuscript.

Dear Authors:

Thank you for revising your manuscript considering all reviewer comments and suggestions. Your manuscript can be published after another careful revision that would be required to improve the following editorial points:

Thanks for the reviewer's positive comment. According to the reviewer's suggestions, we have made the revision in the revised manuscript.

- Line (L) 19 in the track-changed manuscript: Please separate the long (and grammatically incorrect) sentence, like "In addition, photoexposure experiments were designed to compare…".

Thanks for the reviewer's comment, we have made the revision in the revised manuscript.

"In addition, photoexposure experiments were designed to compare photochemical degradation processes of DOM between the SML and the SSW." (Line 19)

- L 26: Please delete the unnecessary phrase "We find that".
- L 27 "is photobleached less": Didn't you use 'photobleaching' in the sense of 'photodegradation'? If it was the case, I think you need to change the phrase as follows (and in the past tense): "was photobleached (photodegraded) more". Please correct me and clarify the sentence, if I misunderstood it.

Thanks for the reviewer's comment, we have made the revision in the revised manuscript.

"Higher EFs were generally observed in the SML in the off-shore regions than in the coastal regions, and CDOM in the SML was photobleached more after relatively strong irradiation, as also indicated by the lower percentages of humic-like DOM and lower specific UV absorbance values ($SUVA_{254}$) in the SML than the subsurface water (SSW)." (Line 26)

- L 31: The concluding sentence is still difficult to follow and hence requires further refinement. Did you mean something like "Our results revealed a new enrichment model for exploring the air-sea interface environment, which can explain the more autochthonous properties of DOM in the SML than the SSW."

Thanks for the reviewer's comment, we have made the revision in the revised manuscript.

"Our results revealed a new enrichment model for exploring the air-sea interface environment, which can explain the more autochthonous properties of DOM in the SML than the SSW." (Line 31)

- L 48-50 "The is a very dynamic interface (Cunliffe et al., 2013), the impact of changes in UV radiation on air-sea fluxes in the SML of important trace gases will need to be assessed.": This and some other sentences are incomplete lacking conjunction. Please conduct a careful proofreading and grammar check-up of the revised manuscript.

Thanks for the reviewer's comment, we have made the revision in the revised manuscript.

"The SML is a very dynamic interface (Cunliffe et al., 2013), moreover, the impact of changes in UV radiation on air-sea fluxes in the SML of important trace gases need to be assessed." (Line 48)

- L 120-122: Repeated dipping was conducted "in the SML up to the depth of 1000 μm (Is this the actual depth for your sampling?)" until the desired volume was collected (11 times, 600 ml).

Thanks for the reviewer's comment, 300 ~ 1000 um is the actual depth for our sampling, and we have made the revision in the revised manuscript.

"Repeated dipping (11 times, 600 ml) was conducted until the desired volume was collected (the depth of the SML sample is nearly 300 ~ 1000 um)." (Line 122)

- L 196-199: Please provide the actual accuracy and precision information, for instance something like % error and CV of repeated reference measurements.

Thanks for the reviewer's comment, we have made the revision in the revised manuscript.

"Two forms of reference water have been developed for DOC analysis. Deep-ocean water, collected at 2600 m in the Sargasso Sea and containing biologically refractory DOC, as well as low carbon water for testing instrument blanks are available to the U.S. and international communities of aquatic chemists (Hansell, 2013; measurement and analytical errors < 19%)." (Line 197)

- L 219: Did you reflect changes in your analytical analyses in this section?

Thanks for the reviewer's suggestion.
This section is the statistical analyses measurement, and we didn't reflect changes in our analytical analyses.

*2.5 Statistical analyses*

The correlation coefficient (R) and probability (P) values were used to evaluate the goodness-of-fit. The correlation matrix, analysis of variance, and principal components analysis were conducted with SPSS version 18.0 (SPSS Inc., Chicago, IL, USA) to determine the possible relationships between the DOM parameters and environmental factors. A P-value ≤ 0.05 was considered significant. Regression analyses between the optical parameters of DOM and several

biogeochemical parameters in the SSW and the SML samples were performed in the Table S1 and

the Table S2, respectively.

- L 309 and Fig. 5 (and other revised figures): Please use 'r' for correlation and 'r2' for regression.

The difference between simple correlation (r) and causal relationship (r2) needs to be checked for

other figures and associated descriptions.

Thanks for the reviewer's comment, we have made the revision in the revised manuscript and we

use 'r' for correlation in Fig. 5.

[Figure]

[Figure]

[Figure]

[Figure]

Fig. 5. Correlations between the microlayer CDOM, DOC, Chl-a, four fluorescence components concentrations, cyanobacteria, phytoplankton biomass, nutrients and bacterial abundance, and their subsurface water concentrations. The dashed lines correspond to the 1:1 lines, and the full lines are the regression models. (All DOM spectroscopic parameters sample were analyzed in spring, summer and winter; Chl-a was determined in spring, summer, and summer; cyanobacteria, phytoplankton biomass, nutrients and bacterial abundance were determined in spring and summer.).

- L 423-427 "All incubation samples were not contaminated, both measurement and analytical errors will let DOC data exceed 100%.": This is an awkward sentence: very difficult to understand its meaning. Please rewrite it.

Thanks for the reviewer's comment, we have deleted this sentence in the revised manuscript.

"Although photodegradation causes CDOM absorption to decrease, DOC is not sensitive to photodegradation in our photodegradation experiments (Fig. 7), implying that the light exposure preferentially removed the colored DOM rather than the non-colored DOM (Bittar et al., 2015)." (Line 424)

- Fig. 3: You still have an r value in one graph. Another issue is the dependent variable: Is salinity dependent or independent variable? If you wanted to explain absorption based on salinity, you need to switch x and y axes. Or if you just wanted to show correlation, please use r instead of r2.

Thanks for the reviewer's comment, we have made the revision in the revised manuscript.

Salinity is the independent variable, hence, we have switched x and y axes.

[Figure]

Fig. 3. Relationships between a(254) and salinity in the subsurface water (SSW) in the East China Sea (ECS) and the Yellow Sea (YS) during spring, summer and winter.